# Physical Exercise as a Therapeutic Approach for Patients Living with Type 2 Diabetes: Does the Explanation Reside in Exerkines?—A Review

**DOI:** 10.3390/ijms26178182

**Published:** 2025-08-23

**Authors:** Daphné Bernard, Ariane Sultan, Karim Bouzakri

**Affiliations:** 1Centre Européen d’Etude du Diabète, Research Unit of Strasbourg University Diabetes and Therapeutics, UR7294, 67200 Strasbourg, France; d.bernard@ceed-diabete.org; 2Diabetology Nutrition Department, Montpellier University Hospital, 34090 Montpellier, France; a-sultan@chu-montpellier.fr; 3Physiology and Experimental Medicine of the Heart and Muscles (Phymedexp) (INSERM, CNRS, Montpellier University Hospital), 34090 Montpellier, France; 4ILONOV, 67200 Strasbourg, France

**Keywords:** type 2 diabetes, exercise, exerkines, insulin sensitivity, beta cell function

## Abstract

For a few decades, Type 2 Diabetes (T2D) has been recognized as a worldwide public health issue. T2D relies on systemic insulin resistance leading to Beta cell dysfunction. Nowadays, lifestyle modifications, such as improving eating habits and increasing physical activity, represent the first recommendations for managing T2D. Physical exercise, as a structured physical activity, is now considered as a non-pharmacological treatment with a wide range of beneficial effects, especially for people living with T2D. The review intends to summarize the current knowledge of physical exercise benefits in a context of T2D: from “unwanted” adipose tissue reduction to Beta cell health improvement. Moreover, we try to suggest an underlying mechanism explaining physical exercise beneficial effects in the context of T2D focusing on exerkines, molecules secreted in response to physical exercise. With this review, we highlight the beneficial impact of post-exercise secretions on Beta cell health and encourage research to continue in this direction. Identifying new exerkines with beneficial effects in the context of T2D could represent a promising approach for managing metabolic diseases.

## 1. Introduction

For a few decades, diabetes mellitus has been recognized as a widespread pandemic. According to the International Diabetes Federation Diabetes Atlas 2025, more than 500 million adults (20–79 years), which represents one in nine individuals worldwide, are affected by the disease. The estimates are alarming, with more than 800 million cases expected by 2050. Diabetes mellitus is characterized by chronic hyperglycemia caused by either Beta cell destruction because of an autoimmune disease, Type 1 diabetes, or a default of insulin action in insulin sensitive tissues, known as insulin resistance, which can lead to Beta cell death. Type 2 diabetes (T2D) is responsible for more than 90% of all types of diabetes [1]. Although many risk factors seem to be attributed to the onset of T2D, obesity and a sedentary lifestyle are considered the leading causes of the disease. Obesity is characterized by an accumulation of fat in the organism, mainly “unwanted” adipose depots surrounding the organs, referred to as ectopic fat. The risk of developing T2D is increased by 4.6-fold for women and 3.5-fold for men when their Body Mass Index (BMI) exceeds 29.9 kg/m^2^ [2]. Industrialization and the rise in Western diet consumption mostly composed of pre-packaged, high-fat, and sweet products have a huge impact on health, including T2D [3]. High calorie intake has demonstrated a negative impact on insulin sensitivity and Beta cell compensation in a cohort of Hispanic women at high risk for T2D [4].

Nowadays, lifestyle modifications, including eating behaviors and increasing physical activity, represent the first recommendations for managing T2D. Physical exercise should be differentiated from physical activity. Physical activity refers to any movement which increases energy use, whereas physical exercise is defined as a structured bout of physical activity. Physical exercise can be categorized according to either resistance or aerobic training. Aerobic training engages large muscle groups inducing energy expenditure. This type of exercise primarily stimulates heart rate with the aim of achieving a high level of endurance performance with cardiovascular system improvement (e.g., jogging, cycling). Resistance training mainly focuses on muscular strength and is designed with various sets of exercises mixed with rest periods (e.g., abdominal crunch, chest press, squats). Combined aerobic and resistance trainings are now more and more described for their promising results [5,6]. Exercise intensity can vary among interventions from low intensity to High Intensity Interval Training (HIIT) and Moderate Intensity Continuous Training (MICT). International guidelines for patients living with T2D recommend at least 150 min/week of moderate to vigorous aerobic exercise with no more than two consecutive days without any physical effort. Moderate to vigorous resistance training should also be added to the exercise routine at least 2 to 3 days per week [6].

Various evidences concluded on the beneficial impact of physical exercise in patients living with T2D [7]. For a few decades, a crosstalk seems to be involved in physical exercise with the secretion molecules defined as “exerkines” [8]. Thus, the aim of this review will be to investigate whether the beneficial effect of physical exercise could come from post-exercise secretions after reviewing the mechanisms involved in the development of the disease and describing the beneficial effects of physical exercise in the context of T2D. Because of the need for regulatory authorities to reduce the use of animals for scientific purposes, we will only present results obtained from studies performed on humans, particularly those with T2D, except in very few examples where data on humans are lacking.

## 2. Type 2 Diabetes Pathophysiology: From Insulin Resistance to Beta Cell Dysfunction

Unhealthy eating habits and/or insufficient physical exercise practice are both leading causes of the onset of T2D. T2D relies on two mechanisms, mainly the development of insulin resistance causing a progressive Beta cell dysfunction.

### 2.1. Insulin Resistance

Insulin resistance consists of an insulin action defect in insulin-sensitive tissues in charge of glucose storage, such as adipose tissues, the liver, and muscles. A hypercaloric diet is responsible for adipose tissue glucose uptake increase in the form of triglyceride, leading to an increase in adipocyte size, referred to as hypertrophy [9]. Depending on their location, adipose tissues play different roles within the body. Subcutaneous adipose tissue, located beneath the skin, stores excess energy as a protective tissue whereas visceral adipose tissue surrounding vital organs is described as deleterious because of its location and its pro-inflammatory state. Hypertrophic adipocytes lead to the development of hypoxia, a default of oxygen supply to the cells. This phenomenon favors the secretion of inflammatory cytokines, negatively impacting the insulin pathway, thus resulting in adipose tissue insulin resistance. Reaching its expandability limit, subcutaneous adipose tissue, as the first caloric buffer, constrains excess energy to be stored in visceral adipose tissue, described to be correlated with insulin resistance. The disruption of insulin inhibitory activity on lipolysis favors the delivery of ectopic lipids to the liver via the portal vein, but also to muscles and the pancreas [10]. Ectopic lipids negatively impact the insulin pathway, resulting in a decrease in muscle glucose uptake and hepatic glycogen synthesis, which contribute to systemic insulin resistance [11].

### 2.2. Beta Cell Dysfunction

Once insulin resistance is settled, a default of insulin action impairs glucose uptake and glucose disposal leading to chronic hyperglycemia. Pancreatic Beta cells will first be able to compensate with an increase in insulin secretion, hyperinsulinemia, until cell death by exhaustion, causing T2D to emerge (Figure 1).

Different compensatory mechanisms enable insulin secretion to be increased. The amount of insulin secreted depends on both the number of Beta cells, which form islets (Beta cell mass), and their capacity to secrete insulin (Beta cell function) [12]. Depending on the organism’s requirements, cells can be remodeled. In patients with insulin-resistance, compared with insulin-sensitive subjects, the surface of the Beta cells is increased by 50% [13]. In response to fatty acids and glucose, the growth factor signaling pathways are activated, through insulin and IGF-I/IGF-II. Phosphorylation of PKB/Akt and IRS2 leads to an increase in the gene expression involved in Beta cell proliferation and cell survival, such as *Pdx-1*. In addition to cellular remodeling, pancreatic islets are also able to compensate through Beta cell functionality improvement. First, insulin biosynthesis can be upregulated through the rise in glucokinase activity, a key enzyme involved in glucose uptake and oxidation, and the transcription of key genes implicated in insulin biosynthesis such as *Pdx-1* [14]. In addition, a positive regulation of insulin secretion can directly respond to insulin resistance and restore glycemia. Thus, it has been suggested that the up-regulation of GLP-1 expression in pancreatic islets could participate in compensation mechanisms by increasing insulin secretion in response to glucose [13].

Although Beta cells can undergo cellular remodeling and induce hyperinsulinemia to counteract insulin resistance, at a certain level apoptosis becomes more pronounced and takes the lead over replication. This leads to a decline in the Beta cell mass [15]. Studies carried out on post-mortem pancreases obtained from individuals who lived with T2D have demonstrated a significant decrease in Beta cell mass compared with lean patients. Alexandra E. Butler et al. have thus revealed a 63% decrease in Beta cell mass in patients living with T2D and overweight, both groups having a similar BMI. This decrease could be mediated by an increase in apoptosis. The frequency of Beta cell apoptosis increased three-fold in patients with T2D and obesity compared with the control group [16].

Nowadays, the first recommendation for patients diagnosed with T2D is an improvement in eating habits combined with a decrease in their sedentary lifestyle.

## 3. Beneficial Impact of Physical Exercise on Type 2 Diabetes

### 3.1. Physical Exercise Improves Glycemic Control in Patients Living with Type 2 Diabetes

Many studies have evaluated the impact of physical exercise on glycemic control in patients living with pre-diabetes or T2D analyzing HbA1c.

First, physical exercise has demonstrated a beneficial impact on HbA1c in patients living with pre-diabetes in a Chinese population. Two types of intervention were studied; patients were either dedicated to aerobic intervention (60–70% maximal heart rate) or a resistance intervention (13 different bouts of resistance exercise per session using an elastic string); both were performed for 6 months, three times a week. Both interventions improved HbA1c compared with the control group, with no significant difference between aerobic and resistance training [17].

In patients at a more advanced stage of the disease, living with T2D, aerobic exercise intervention of at least 12 weeks revealed a significant improvement in glycemic control [18], with no intensity dose response for a physical effort ranging between 30 and 80% of maximal exercise capacity [19]. Jansson, A.K et al. have also demonstrated a beneficial impact of resistance exercise, compared with a control group, on HbA1c [20]. A meta-analysis gathering results from 158 clinical trials and performed for a period of at least 12 weeks revealed that HIIT induced the most noticeable reduction in HbA1c, followed by combined training, and aerobic or resistance training alone [21]. Nowadays combined exercise is studied more and more because of its promising effect on the disease. Although a long period of combined exercise (>8 weeks) [22,23] or a small amount of combined exercise (1 week) can enhance glycemic control [24], some conflicting data were reported in a meta-analysis evaluating HbA1c in women living with T2D after combined training [25].

### 3.2. Physical Exercise Improves Insulin Sensitivity in Patients Living with Type 2 Diabetes

Petersen, M.H. et al. have demonstrated a beneficial impact of an 8-week supervised HIIT program, combining rowing and cycling (3 sessions per week), in patients living with T2D on insulin sensitivity assessed by a hyperinsulinemic euglycemic clamp. Their Insulin-stimulated Glucose Disposal Rate (GDR) increased by 42% after the intervention [26]. Christ-Roberts, C.Y. et al. also reported an improvement of insulin stimulated GDR in patients with T2D after exercising for 8 weeks by gradually increasing intensity (from 60% at the beginning of the study to 70% VO_2_ max at the end of the intervention), duration (from 20 min at the beginning of the study to 45 min at the end of the intervention), and frequency (three times a week at the beginning of the study to four times a week at the end of the intervention) [27]. A moderate and high exercise dose group has reported an improvement in the late-phase glucose-stimulated Insulin Sensitivity Index (ISI) [28]. A meta-analysis gathering results from studies involving aerobic and resistance interventions performed for at least 12 weeks revealed insulin sensitivity improvement in patients living with T2D, which can persist for up to 72 h after the last bout of effort [29]. Homeostatic Model Assessment of Insulin Resistance (HOMA-IR), a mathematical model representing insulin resistance state in individuals, can also evaluate insulin sensitivity. HOMA-IR is obtained from fasting plasma glucose and insulin [30]. Combined aerobic and resistance exercise can have a beneficial impact on insulin resistance, decreasing HOMA-IR in post-menopausal women with T2D [25] or in cohort with no gender distinction [31]. Beyond systemic plasmatic insulin resistance measurement, evaluating insulin sensitivity post-exercise in peripheral organs where insulin resistance may occur should also be interesting. For this purpose, Gregory, J.M. et al. have demonstrated that an aerobic exercise realized during approximately 15 weeks decreases liver endogenous glucose production after insulin stimulation and improves muscle insulin sensitivity [32]. These results can be explained by the impact of physical exercise on a molecular level. Thus, Hussey, S.E. et al. have reported an increase in GLUT4 expression in adipose tissues (+36%) and muscle biopsies (+20%) after a 4-week combined moderate and high intensity exercise according to the following protocol: MICT intervention: 3 days a week at 60% Wmax and HIIIT intervention: 2 days a week with 6 × 5 min bouts of effort performed at 70% Wmax [33]. Two other studies have evaluated a beneficial impact of exercise on GLUT4 protein expression but without any impact on the insulin pathway [27,34].

### 3.3. Physical Exercise Decreases Ectopic Lipids and Visceral Adipose Tissue in Patients Living with Type 2 Diabetes

Visceral adipose tissue surrounding vital organs, or ectopic lipids mainly located around the liver, pancreas or into muscle fibers are known to be deleterious for the organism, which can lead to T2D. Ectopic lipids are associated with insulin resistance, glycemia dysregulation and an increasing risk of developing cardiovascular diseases [35]. Thus, a decrease in ectopic lipids and visceral adipose tissue could be beneficial for patients living with T2D.

Cassidy, S. et al. have already demonstrated a loss of 39% liver fat mass after a HIIT effort for 12 weeks in patients living with T2D correlated with a reduction in HbA1c [36]. Sabag, A. et al. have also concluded on a decrease in liver fat percentage in patients with T2D after bouts of both HIIT (4 min of cycling at 90% VO_2_ max + 10 min warm up and 5 min cool-down) and MICT (training duration ranging from 30 min to 45 min by the 4th week and physical effort performed at 60% VO_2_ max) exercise during a period of 12 weeks without any significant difference in the type of exercise interventions [37]. Resistance exercise has also demonstrated encouraging results for liver fat reduction. Liver fat mass was decreased after a home-based resistance exercise for 6 months, three times a week, in patients living with T2D [38]. For Freer, C.L. et al., resistance training seems to be effective as weight loss without any additional impact of exercise on the fatty liver index [39]. Pancreatic fat is also deleterious for Beta cell functionality [40]. Li, M. et al. described a decrease in pancreatic fat after a 6-month moderate-intensity aerobic exercise (three times a week—60–70% maximal heart rate) [41]. A shorter intervention (2 weeks) also induces a reduction in pancreatic fat in patients living with pre-diabetes and T2D diabetes for both moderate (MICT: 60% of peak workload) and high (Sprint Interval Training: supramaximal workload) intensities [42]. Beyond liver and pancreas fat, Intermuscular Adipose Tissue (IMAT) is also considered as an ectopic fat depot associated with insulin resistance [43] and T2D [44,45]. Tang, F. et al. have shown that a resistance exercise performed for 6 months revealed a decrease in 9.89% IMAT in patients living with T2D [46]. The Look AHEAD (Action for Health in Diabetes) trial has also demonstrated a beneficial impact of an exercise intervention in patients with T2D. A moderate intensity aerobic effort realized in routine for one year has prevented IMAT increase [47].

Compared with subcutaneous adipose tissue described to have a protective effect on health, visceral adipose tissue increases the risk to develop insulin resistance [48]. Thus, a decrease in visceral fat seems to be beneficial for people living with T2D. A meta-analysis evaluating 24 studies has reported a beneficial impact of aerobic exercise only on visceral fat reduction [35]. Kazeminasab, F. et al. have also noticed a reduction in visceral fat for aerobic, resistance, and combined training [49]. These results confirm the beneficial impact of a training intervention on visceral adipose tissue but emphasize the need to clarify the impact of each type of exercise.

### 3.4. Physical Exercise Enhances Beta Cell Mass in Patients Living with T2D

Hyperglycemia caused by insulin resistance leads to Beta cell compensation that exacerbates insulin secretion resulting in Beta cell exhaustion and apoptosis. Maintaining the Beta cell mass is therefore essential for T2D remission. Human Beta cells exposed to plasma collected from non-diabetic patients after exercising decreased inflammatory cytokines (IL1-β and IFN-γ) and apoptosis markers [50]. Interestingly, Coomans de Brachène, A. et al. have evaluated serum obtained post exercise from patients with T2D to assess its impact on Beta cell apoptosis. The intervention consisted of high intensity physical effort combined with strength training for 12 weeks. They demonstrated a 26% reduction in apoptosis on EndoC-βH1, a Beta cell line, treated for 24 h with serum obtained from exercised patients with T2D [51].

Several studies have investigated the impact of physical exercise on the Beta cell mass in rodents [52,53,54,55,56], but only few studies focused on humans [50,51]. The enhancement of Beta cell mass could rely on an increase in Beta cell proliferation or Beta cell apoptosis reduction [57]. More studies are needed to understand the impact of physical exercise on Beta cell mass and the underlying mechanism in the context of T2D.

### 3.5. Physical Exercise Enhances Beta Cell Functionality in Patients Living with Type 2 Diabetes

Despite the importance of Beta cell mass in supplying insulin requirements, Beta cell functionality, beyond number of cells, is essential for sustaining glucose homeostasis. Beta cell functionality refers to the cell’s capacity to secrete insulin in response to glucose variations. The impact of physical exercise on Beta cell functionality can be assessed with its ability to sense glucose and secrete insulin [57].

First, Beta cell ability to sense glucose is crucial for glucose homeostasis. In a post-prandial state, Beta cells respond to glucose variations with glucose uptake via glucose transporters, GLUT1 in human and GLUT2 in rodents. Intracellularly, glucose is metabolized in glycolysis through different enzymes, mainly glucokinase. A membrane depolarization operates in response to a rise in intracellular concentration because of a change in ATP/ADP ratio and Ca^2+^ closing channels resulting in K^+^ opening channels, thus inducing insulin secretion mediated by insulin granule exocytosis [58]. Several studies investigated the impact of physical exercise on Beta cell glucose sensing, but only in rodents [57], probably due to the complexity to obtain human pancreatic Beta cells from cadaveric donors. Király, M.A. have demonstrated an increase in GLUT2 protein staining in pancreatic sections collected from Zucker Diabetic Fatty rats after realizing a swimming exercise program for 3 months [59]. An increase in glucokinase expression was also reported by Choi, S.B. et al. in Male Sprague-Dawley pancreatectomized rats, after exercising in an uphill treadmill for 30 min four times a week during 8 weeks [60].

Other factors can also reflect Beta cell functionality, notably HOMA-β calculated with fasting glucose, fasting insulin, and the Disposition Index (DI), measured according to insulin secretion adjusted with insulin sensitivity [61]. Insulin sensitivity should be considered when studying Beta cell functionality with insulin secretion knowing that, if insulin resistance increases, Beta cells will compensate by secreting more insulin without reflecting Beta cell health [62]. An intensive lifestyle intervention consisting of combined (aerobic + resistance) training associated with a dietary intervention in patients living with T2D has demonstrated a beneficial impact on Beta cell functionality measured with a 40% increase in the DI compared with that of the standard care group [63]. A moderate (three times a week: two aerobic training sessions and one combined training session) or high volume (six times a week: four aerobic training sessions and two combined training sessions) exercise associated with a calorie restriction shows a better improvement of the DI compared with the control group [28]. Exercise alone can also improve Beta cell functionality in patients with T2D as a 10-week moderate-intensity exercise performed up to four times a week has demonstrated a 38% increase in late-phase DI [64]. A 6-month resistance exercise training performed by individuals with T2D also revealed an improvement in Beta cell functionality measured with HOMA2-β [46]. An 8-week supervised HIIT program combining rowing and cycling in people living with T2D described an improvement in the DI of up to 200%, depending on individuals, but did not restore Beta cell functionality, if a comparison was made with the control group [26]. In addition, an aerobic training coupled with medication such as the GLP1R agonist Semaglutide can also improve insulin secretion in people with T2D compared with the training group alone, with a seven times greater effect [65]. In summary, studies analyzing the impact of physical exercise on insulin secretion seem to have demonstrated the need to exercise for at least 2 months to achieve positive results on Beta cell functionality [62]. Physical exercise benefits are summarized in Figure 2.

There is a large amount of evidence available to validate the beneficial impact of physical exercise on metabolic parameters for patients living with T2D in a systemic approach. However, the exact mechanisms underlying these results need to be clarified. Nowadays, a great interest is shown on inter-organ crosstalk, especially exercise-mediated secretion, referred to as exerkines [66].

## 4. Physical Exercise’s Beneficial Impact Could Rely on Endocrine Secretions

As seen, physical exercise is widely described as a beneficial non-pharmacological treatment to regulate metabolism and therapy for metabolic diseases such as T2D as it can improve glucose homeostasis. Moreover, physical exercise possesses an anti-inflammatory role as it can decrease pro-inflammatory cytokines, a vicious cycle involved in patients living with T2D [67,68]. Nowadays, the aim is to better understand the mechanisms underlying such a positive impact. For a few decades, the role of exerkines, referring to a “signaling molecule that is released in response to acute and/or chronic exercise stimuli and exerts its effects through endocrine, paracrine, and/or autocrine pathways” [69], emerges as the potential explanation. Several tissues, in response to exercise stimuli can secrete exerkines, such as skeletal muscles, adipose tissues, and the liver, which represents more than a hundred or thousand proteins [66]. The impact of physical exercise on exerkine concentration and their potential effects are summarized in Table 1 and in Figure 3.

### 4.1. Impact of Physical Exercise on Exerkine Secretion in Patients with Type 2 Diabetes

#### 4.1.1. Skeletal Muscle Secretions

Beyond its contractile function during exercise, the skeletal muscle is considered one of the most important endocrine organs, enabling the secretion of various biologically active molecules defined as myokines [69].

Interleukin-6 (IL-6) was the first exerkine identified by Steensberg, A. et al. in the 2000s [70], paving the way for the emergence of a multitude of exercise-stimulated proteins. IL-6 is a pleiotropic cytokine since it is secreted by different tissues such as muscle and adipose tissue, or more broadly by all multinucleated cells. IL-6 presents a dual function because of its pro- or anti-inflammatory role depending on the physiological context. IL-6 seems to be implicated in insulin resistance as its plasma concentration is often increased in patients with T2D. In a paradoxical way, IL-6 is also released in blood circulation in response to physical exercise and seems to have an anti-inflammatory effect by inhibiting the effect of tumor necrosis factor α (TNF-α) or stimulating the release of anti-inflammatory cytokines such as Interleukin-10 (IL-10) [71]. Carey, A.L. et al. have reported an increase in insulin-mediated glucose disposal in response to an IL-6 infusion in healthy men. This effect seems to be mediated by AMPK activation [72]. Although IL-6 seems to be related to the beneficial impact of physical exercise with an increase in its circulatory concentration [73], conflicting results were reported by describing a reduction in pro-inflammatory cytokines, such as IL-6, through two different meta-analyses of randomized controlled trials [74,75]. More data are needed to understand the effective role of IL-6 in physical exercise, probably due to its dual facet. Heterogeneity in physical exercise modalities could explain these contradictory results.

Irisin is another well-known myokine described for the first time in 2012 by Pontus Boström et al. It results from the cleavage of fibronectin type III domain-containing protein 5. Irisin is regulated by PGC1α and is described to be implicated in thermogenic functions in adipose tissues by upregulating UCP1 expression [76]. Adipose tissues represent a highly dynamic tissue because of its implication in different metabolic functions. Two main adipose depots are identified according to their own functions and morphology. White adipose tissues store excess fatty acids in the form of triglycerides in lipid droplets and can sustain energy demand by releasing them. Brown adipose tissues (BAT) participate in thermogenesis, the capacity to generate heat from substrates and maintain optimal body temperature. During adulthood, the development of another adipose cell type, identified as beige adipocytes and derived from white adipocytes with a BAT-like phenotype, represents a promising therapeutic strategy for people with T2D due to its involvement in glucose homeostasis and lipid metabolism [77]. Despite being involved in adipose tissues with thermogenic functions, irisin is also described as a pleiotropic molecule with multi-organ effects such as its role in protection cognitive functions [78]. Different meta-analyses have reported lower levels of irisin in patients living with T2D compared with the control group [79,80,81]. It can thus be postulated that a decrease in irisin concentration in people with T2D can be implicated in the progression of the disease. Several studies have thus investigated whether physical exercise can restore irisin levels. Motahari Rad, M. et al. reported an increase in irisin concentration after a 12-week training session of combined exercise (aerobic and resistance) in men with T2D [82]. A systematic review, including six studies performed in patients living with T2D who underwent exercise training for at least four weeks, has investigated irisin response after exercise. One study did not reveal any impact of physical exercise on irisin concentration. However, four studies reported a significant increase in the blood irisin level in the intervention group, with a variation (baseline versus post-intervention) of +39% compared with the control group (−5%). To date, there is still insufficient evidence to define which exercise modality (aerobic or exercise) induces a greater increase in irisin concentration. Regarding intensity, high intensity effort seems to reflect a better positive impact [83]. Moreover, Villamil-Parra, W. et al. also reported an increase in irisin concentration after physical exercise in patients with metabolic syndrome, not exclusively T2D [84].

Brain-derived neurotrophic factor (BDNF) acts in the central and peripheral systems as a growth factor to stimulate neuron proliferation and protection and was isolated for the first time in 1989 by Yves-Alain Barde and Hans Thoenen [85]. Despite its role in the brain microenvironment, BDNF is not exclusively synthesized in the brain but also in muscle tissues. It is thus a muscle-secreted molecule, especially in response to a physical-exercise stimulus. It has a broad range of activities because of its biologically active isoforms and its different receptors implicated in several signaling pathways [85]. In T2D, BDNF has been described to have metabolic effects in peripheral tissues (liver, muscle, adipose tissues, and pancreas) and could participate in insulin sensitivity improvement through a reduction in food intake, the prevention of pancreas cell death, or by decreasing glucose and insulin levels [86,87]. An insulin resistance state or T2D profile are associated with a low concentration of BDNF [88]. Thus, an increase in BDNF levels could be favorable to alleviate metabolic outcome in patients living with T2D. Several studies have, thus, investigated whether physical exercise can restore BDNF levels. First, Villamil-Parra, W. et al. demonstrated that physical exercise not only increases irisin blood levels but also increases BDNF in patients with metabolic syndrome. Its effect seems to be linked to a positive effect on mental health [84]. However, the impact of physical exercise on BDNF blood concentration seems to be conflicted, as described by Jamali, A. et al. in a systematic review including human and animal studies. In 11 of the studies included, 5 articles demonstrated an increase in BDNF concentration, 4 a reduction, and 2 articles reported no differences [88]. Further studies are needed to confirm or deny the impact of physical exercise on circulating BDNF level.

Myostatin, another myokine secreted in response to physical exercise, is known to be implicated in metabolic syndromes such as obesity or diabetes. This secreted growth factor decreases skeletal muscle mass [89]. Hjorth, M. et al. have demonstrated a negative correlation between myostatin expression and insulin sensitivity in skeletal muscles [90]. Moreover, results from Brandt, C. et al. have established an increase in myostatin mRNA expression in patients with T2D compared with control subjects. However, plasma myostatin concentration did not suggest an implication of T2D [91]. Various studies have investigated the impact of physical exercise on myostatin levels in patients with T2D. First, an aerobic training carried out at a moderate intensity for 6 months on 10 insulin-resistant men have shown a decrease in myostatin protein and plasma levels [92]. Furthermore, an aerobic training performed for 12 weeks, three times a week, either for a high intensity (HIIT) or moderate intensity (MICT) intervention revealed a decrease in myostatin plasma level 48 h after the last intervention session in adult men with T2D [93]. A resistance training performed for 12 weeks, 3 days a week, also demonstrated a reduction in myostatin as suggested by Shabkhiz, F. et al., while the myostatin baseline was higher in men with T2D compared with the group without T2D. Thus, a decrease in myostatin levels induced by physical exercise could alleviate insulin resistance [94]. However, all these findings were reported in men with T2D. More data are needed with no gender distinction.

#### 4.1.2. Adipose Tissue Secretions

Adiponectin is a well-known adipokine involved in different metabolism processes. Circulating adiponectin levels are negatively associated with diabetes and pre-diabetes development especially in individuals with few T2D risk factors [95]. In addition, higher levels of circulating adiponectin have been reported to be associated with a lower risk of T2D [96]. Many studies have already investigated the impact of physical exercise on adiponectin in patients with established T2D. A recent published meta-analysis revealed a significant increase in the adipokine after physical exercise [75]. Additionally, a high intensity effort or an aerobic + resistance training performed for 12 weeks in patients living with T2D enhances adiponectin concentration [97]. Blüher, M. et al., in a cross-sectional study, also reported a beneficial impact of an aerobic training (20 min of running/biking, 20 min of swimming, 20 min of warming up and cooling down) conducted over a 4-week period, three times a week on circulating adiponectin [98]. The wide range of adiponectin effects in the context of insulin resistance prevention mainly relies on AMPK activation [99]. Adiponectin also demonstrates a beneficial impact on pancreatic Beta cells through glucose-stimulated insulin secretion, a reduction in Beta cell death, or increased Beta cell survival [100]. Adiponectin is also known to play a positive role in glucose uptake in peripheral tissues such as adipose tissues. Thus, the beneficial effect mediated by physical exercise on adiponectin concentration may suggest a therapeutic approach for patients with T2D [101].

Apelin was first identified in 1998 [102]. Although apelin is initially described as an adipokine, it is also be secreted by other tissues such as muscles, the kidneys, and the heart [103,104]. Some studies have reported a decrease in circulatory apelin in patients with newly T2D diagnosis [105], whereas others have suggested in a meta-analysis an increase in apelin concentration in T2D profiles versus lean controls [106]. Although, data on apelin plasma level in patients with T2D compared with lean subjects seem contradictory, the beneficial impacts of apelin are now well established: apelin enhances glucose uptake in muscle cells and improves insulin sensitivity thereby attenuating inflammation-driven insulin resistance [69]. However, despite the beneficial role of apelin in metabolic homeostasis, apelin concentration seems to be correlated with insulin resistance, probably explained by a compensatory effect [104,107]. Aerobic exercise realized four times a week, with each session lasting 60 min, by patients living with type 2 diabetes for 12 weeks (60–75% maximal heart rate) supports an increase in apelin secretion, but this effect was only reported in women [108]. Moreover, a meta-analysis investigating the impact of physical exercise on exerkine concentrations in patients living with T2D has found three studies studying apelin for an intervention of at least 2 weeks. However, they do not report any modification in apelin concentration in comparison with the control group [75].

Compared with adiponectin and apelin, described as beneficial adipokines, others, resistin and visfatin, are more deleterious in the context of insulin resistance, while leptin has a more controversial effect. Although leptin is known to be beneficial in healthy individuals because of its role in satiety and cellular metabolism, an obese state could lead to leptin resistance and hyperleptinemia, resulting in a decrease in its action, exacerbating the pathogenicity of the disease. Leptin levels are associated with T2D macro- and microvascular complications [109]. As its name suggests, resistin is implicated in insulin resistance development. Its secretion is associated with inflammation and could then disrupt the insulin pathway [110]. Finally, visfatin is mainly expressed in visceral adipose tissues and is positively correlated with insulin resistance [111]. Visfatin is also associated with the risk of developing atherosclerosis in T2D [112]. Thus, a decrease in these three adipocytokines could be relevant for T2D management. García-Hermoso, A. et al. have reported in a recent meta-analysis published in 2023, that 10, 4 and 3 studies, respectively studying leptin, resistin, and visfatin that all the three adipokines were reduced after physical exercise [75]. These results are in accordance with Becic, T. et al.’s meta-analysis which also reported a reduction in leptin level in patients with prediabetes and T2D [113]. Kadoglou, N.P. et al. demonstrated a decrease in the circulating resistin level in individuals with T2D after realizing a 16-week aerobic training intervention, four times a week, with each session lasting 45–60 min at 85% VO_2_ max [114], whereas others suggested no significant change [115,116].

#### 4.1.3. Liver Secretions

Beyond skeletal muscle and adipose tissue, the liver is also considered an endocrine organ, and physical exercise can result in hepatokine secretion [117].

Fibroblast Growth Factor-21 (FGF-21) is a liver-secreted cytokine involved in glucose homeostasis [117]. Several studies have demonstrated a beneficial impact of FGF-21 analogs in dyslipidemia in people living with T2D and obesity. FGF-21 can also alleviate inflammation and improve lipid and glucose metabolism [75]. However, an up-regulation of FGF-21 in patients living with T2D has been reported and correlates with hepatic and muscle insulin resistance, making it a potential early biomarker of the disease [118,119]. In a meta-analysis performed by García-Hermoso, A. et al., six studies have investigated the role of physical exercise in patients with T2D on FGF-21 concentration and have concluded a significant increase in FGF-21 [75]. In line with these results, Sabaratnam, R. et al. have reported an up-regulation of muscle *FGF-21* mRNA expression after a 1 h-bout of exercise in men with T2D. Plasma FGF-21 was only slightly increased and returned to baseline levels 3 h after [120]. However, a study analyzing the impact of combined exercise on different serum cytokine concentrations has reported no change in FGF-21 compared with the baseline [82]. A study performed by Hansen, J.S. et al. suggests the implication of pancreas-secreted hormones (glucagon and insulin) in FGF-21 regulation in a context of physical exercise as secretions are modulated with a pancreatic clamp [121]. An increase in glucagon/insulin ratio during physical exercise seems to be the main activator of FGF-21 production. As a potential therapeutic approach, physical exercise could be a promising non-pharmacological treatment to increase FGF-21 concentration, although T2D seems to impair physical exercise response [122].

Fetuin-A is an hepatokine involved in metabolic diseases such as insulin resistance, liver fibrosis, and T2D [117]. A higher fetuin-A plasma level is associated with an increasing risk of developing T2D and is even more important in individuals with high glucose levels [123,124]. García-Hermoso, A. et al., in their meta-analysis based on four studies investigating the impact of physical exercise in T2D patients, have noted a significant decrease in fetuin-A after the intervention [75]. Results from Ramírez-Vélez, R. et al. meta-analysis are more mitigated. They described a reduction in fetuin-A levels after a supervised exercise in a large population of people with obesity and T2D, but a sub-group analysis does not seem to conclude on an impact in patients with T2D only [125]. A reduction in an average of 11% of fetuin-A plasma concentration seems, after performing a 12-week combined exercise, to be associated with a decrease in inflammation markers. Although this study was not investigated in patients with T2D, it can give some evidence on the mechanisms underlying fetuin-A post-exercise modification. Physical exercise-mediated insulin sensitivity improvement could be partially explained by a change in circulatory fetuin-A associated with a decrease in FFA leading to a reduction in inflammation signaling [126].

Follistatin is mainly secreted by the liver and, as well as for FGF-21, its production is activated by a rise in glucagon-to-insulin level [121,127]. As for fetuin-A, a high level of follistatin is associated with an increasing risk of T2D [128]. The hepatokine is involved in glucose metabolism, most specifically it participates in metabolic disorders [129]. Studies investigated the role of physical exercise on fetuin-A in individuals living with T2D are really few. Only one was reported in García-Hermoso, A. et al.’s meta-analysis [75]. Thus, Motahari Rad, M. et al. noticed a significant increase after realizing a combined exercise protocol (aerobic and resistance training) for 12 weeks with three sessions per week [82]. More data is needed to evaluate the real impact of physical exercise on follistatin and if it participates to the exercise overall beneficial impact on metabolic parameters in patients with T2D.

**Table 1 ijms-26-08182-t001:** Impact of physical exercise on exerkine levels and their potential effects. Abbreviation: =: no change; ↗: increase; ↘: decrease; Aex: Aerobic training; BDNF: Brain Derived Neurotrophic Factor; CVD: Cardiovascular Disease; CT: Combined training; FGF-21: Fibroblast Growth Factor-21; HIIT: High Intensity Interval training; IGT: Impaired glucose tolerance; IL-6: Interleukin-6; IR: Insulin Resistance; MA: Meta-analysis; MS: Metabolic Syndrome; NA: not available; NAFLD: Non-alcoholic fatty liver disease; NGT: Normal Glucose Tolerance; PM: Post-menopausal; RCT: Randomized controlled trials; RT: Resistance training; SR: Systematic Review; w: with; w/o: without.

Exerkines	Studies	Patients	Type of Intervention	Physical Exercise Impact
** Muscle Secretions **				
IL-6	[74]: MA RCT	T2D (w or w/o obesity, CAD, overweight)	Aex, RT, CT	↘
	[75]: MA RCT	T2D	Aex, HIIT, RT, CT	↘
Irisin	[82]: RCT	T2D (men)	CT (Aex + RT or RT + Aex)	↗
	[84]: SR	MS	NA	↗
	[83]: SR RCT	T2D	Aex, RT, HIT, MIT	4 ↗ and 1 =
BDNF	[88]: SR RCT	T2D (4 humans and 7 animals)	Aex (humans)	5 ↗; 4 ↘ and 2 =
Myostatin	[92]	IR (men)	Aex	↘
	[93]: RCT	T2D (men)	Aex (HIIT vs. MICT)	↘
	[94]: RCT	T2D (elderly men)	RT	↘
** Adipose Tissue Secretion **				
Adiponectin	[75]: MA RCT	T2D	Aex, RT, CT, HIIT	↗
	[97]: RCT	T2D + MS	Aex, CT	↗
	[98]	IGT, T2D	Aex	↗
Apelin	[108]: RCT	T2D + overweight	AEX	↗
	[75]: MA RCT	T2D	Aex, RT, CT, HIIT	=
Leptin	[75]: MA RCT	T2D	Aex, RT, CT, HIIT	↘
	[113]: MA RCT	T2D and pre-diabetes	Aex, RT, CT	↘
Resistin	[75]: MA RCT	T2D	Aex, RT, CT, HIIT	↘
	[114]: RCT	T2D + overweight/obese	Aex	↘
	[115]: RCT	T2D (PM women)	Aex + diet intervention	=
	[116]	NGT, IGT, T2D	Cardiovascular exercise + caloric restriction	=
Visfatin	[75]: MA RCT	T2D	Aex, RT, CT, HIIT	↘
** Liver Secretions **				
FGF-21	[75]: MA RCT	T2D	Aex, RT, CT, HIIT	↗
	[120]	T2D (men) + overweight/obese	Aex	↗
	[82]: RCT	T2D (men)	CT (Aex + RT or RT + Aex)	=
Fetuin-A	[75]: MA RCT	T2D	Aex, RT, CT, HIIT	↘
	[125]: MA RCT	T2D, obese, NAFLD, CVD	Aex, CT	obese ↘ but T2D =
Follistatin	[82]: RCT	T2D (men)	CT (Aex + RT or RT + Aex)	↗

### 4.2. Impact of Exercise-Induced Secretions on Pancreatic Islet Health

When studying the beneficial effect of physical exercise on metabolic diseases such as T2D, its impact on Beta cell health should be one of the main indicators to be considered. However, only a few studies are studying the direct impact of exerkines on pancreatic islets.

#### 4.2.1. Impact of Post-Exercise Plasma on Pancreatic Islet Health

To our knowledge, only one paper has studied the direct impact of post-exercise plasma originating from patients with T2D. Thus, Coomans de Brachène, A. et al. have investigated the impact of post-exercise plasma collected from patients living with T2D after having followed a 12-week intervention program. This study has highlighted the protective effect of trained plasma on Beta cells from thapsigargin-induced apoptosis with a 26% decrease in apoptotic cells associated with a reduction in some pro-apoptotic genes such as *CHOP*, *XBP1,* and *DP5* mRNA expression. In addition, they have tested the impact of an anti-inflammatory cytokine, clusterin, and assessed whether it can reproduce the beneficial effect of physical exercise. They have demonstrated that clusterin can reduce thapsigargin-induced cell death by 31–42% at two concentrations (1 and 100 ng/mL) [51]. This result can suggest a beneficial impact of physical exercise mediated by exerkines. Others have treated Beta cells with plasma collected either from non-diabetic individuals [50] or from a rodent’s diabetic model [53] and concluded a reduction in plasma cytokines (IL1-β and IFN-γ) or stress factors responsible for apoptosis or an increase in Beta cell proliferation, respectively.

#### 4.2.2. Impact of Exerkines on Human Pancreatic Islet Health

Furthermore, several papers have investigated the direct impact of some exerkines on Beta cell health [130]. Because the procurement of T2D human Beta cells is challenging, we will investigate the overall impact of exerkines in healthy human pancreatic Beta cells. First, the recombinant myokine irisin has been shown to enhance glucose-stimulated insulin secretion in human pancreatic Beta cells and increase Insulin (*Ins*) mRNA expression. In addition, irisin can also reverse palmitate-induced Beta cell apoptosis and increase proliferation [131].

Rutti, S. et al. have investigated the impact of angiogenin and osteoprotegerin, tricep-specific myokines, in muscle–pancreas crosstalk. They have demonstrated an anti-apoptotic effect of both myokines on human Beta cells [132]. Although osteoprotegerin is secreted by muscles, its primary source appears to be endothelial cells and adipocytes [133], while angiogenin is described to be mainly secreted by endothelial and muscle cells [134]. Angiogenin possesses a broad spectrum of actions and is mainly involved in angiogenesis and neuroprotection [135]. Osteoprotegerin also participates in neovascularization [136]. These two myokines seem to be implicated in micro- and macro-vascular complications related to T2D [137,138,139]. Conflicting results were found for T2D influence on angiogenin levels. Some report a lower concentration of the exerkine in a cohort of patients with T2D [137,139], while others suggest an up-regulation and an association with cardiovascular risks [138]. When patients suffer from poorly controlled diabetes, low angiogenin concentrations have been described to be associated with a longer duration of the disease [139]. Thus, an increase in angiogenin levels found in some studies in patients living with T2D could be partially explained by a compensatory mechanism. Conversely, for osteoprotegerin, results seem to converge toward an increase in plasma levels in patients with T2D compared with a healthy group [133,140,141,142]. Depending on the secretory context, secreted molecules can reveal a dual facet. It can be argued that physical exercise could thus beneficially alter their functions and induces a promising approach for Beta cell health.

Less studied is decorin, a myokine secreted by myotubes in response to contraction. Its plasma concentration is up regulated after a resistance exercise bout in humans and is described to be implicated in muscle hypertrophy [143]. Decorin has been described to improve insulin secretion in pseudo-islets. Pseudo-islets treated with decorin at 50 µg/mL induces a 58% increase in insulin secretion after glucose stimulation (20 mM) and a 47% increase in the stimulation index [144]. A study from our team has investigated the impact of a decorin treatment on human Beta cells from healthy and T2D donors and have concluded an increase in glucose-stimulated insulin secretion in Beta cells isolated from T2D donors associated with an increase in insulin content [145]. Decorin is a ubiquitously expressed proteoglycan, constitutive of the extracellular matrix. Its biological functions cover a wide range of activities such as collagen fibrillogenesis, angiogenesis, innate immunity, or inflammation enabled by its affinity with multiple receptors by activation or inhibition [146]. Moreover, proteoglycan participates in glucose regulation as demonstrated in decorin-deficient mice. A loss of decorin induces a reduction in glucose tolerance [147]. A decorin treatment can also have a cardioprotective effect has demonstrated in a T2D rodent model, but in this case, the beneficial impact of decorin does not seem to be associated with an improvement in glucose tolerance [148]. Plasma decorin levels have been described to be higher in patients with T2D compared with a normal glucose tolerant control group, probably due to a compensatory mechanism [149]. Decorin secretion in response to physical exercise can thus present a promising approach for managing T2D, most specifically by targeting Beta cell health.

In addition, fractalkine has been described to be modulated in response to exercise and is thus considered as an exerkine [8,150]. Lee, Y.S. et al. have demonstrated a positive effect of fractalkine on glucose stimulated insulin secretion (+58%) but preferentially mediated by an indirect mediator rather than acting as an insulin secretagogue [151]. These results are in accordance with those from Rutti et al. who have also demonstrated a protective role for the exerkine in Beta cells [152]. Fractalkine is expressed by a wide range of cell types such as endothelial cells or neurons in a physiological state and has also been described to be secreted by adipocytes [153]. Its first role was attributed to neuroprotective functions after brain injuries. As a chemokine, fractalkine is also considered an adhesion molecule in charge of cell immunity recruitment. In a pathological context, fractalkine can also be implicated in more adverse effects [154]. The chemokine is also involved in metabolic processes. Fractalkine knockout mice exhibit an impaired glucose tolerance while fractalkine administration has been reported to restore glycemia by enhancing insulin secretion [151]. It has been reported that fractalkine levels were elevated in patients with T2D compared with a control group [153,155] and is associated with inflammation [156]. Thus, once again, depending on the secretory context, secreted molecules can reveal a dual facet. Physical exercise could thus beneficially alter their functions for greater impact, notably mediated for Beta cell health improvement.

## 5. Conclusions

Our review aimed to summarize current studies on the beneficial impact of physical exercise in patients living with T2D. Exercise reduces visceral adipose tissues and ectopic lipids thereby helping to reduce insulin resistance. In addition, by enhancing insulin-sensitivity, it promotes glucose uptake in peripheral tissues and participates in glucose homeostasis, measured by a control HbA1c. Reducing hyperglycemia could then restore Beta cell functionality and Beta cell mass. Cytokines secreted in response to physical exercise could mediate the positive role of physical efforts. Indeed, various promising effects have been reported such as glucose homeostasis and insulin sensitivity improvement, a reduction in the pro-inflammatory state, and the induction of browning. Browning activation has been suggested to be a promising therapeutic approach in patients living with T2D [157]. Although, less is known about the impact of exerkines on body composition, some evidences suggest a role of these molecules secreted in response to exercise on lipid metabolism [158]. Beta cells directly treated with either plasma collected after exercising or exerkines have demonstrated a protective effect against apoptosis and an improvement of glucose-stimulated insulin secretion. All these findings highlight the need to go further in understanding the hidden side of physical exercise and the associated cellular and molecular mechanisms, especially in human studies.

Although physical exercise has demonstrated a beneficial impact in patients living with T2D, the diversity of intervention protocols makes it difficult to suggest the most appropriate program for patients with T2D. In addition, heterogeneity found in the results could be attributed to individuals’ variability [159]. A sub-group analysis performed in a study investigating changes in exerkine concentrations revealed no difference in effects according to the type of exercise. However, a session duration of at least 60 min appears to be more effective than sessions lasting less than 60 min [75]. The most suitable period of the day to exercise has yet to be determined [159]. Thus, more data are needed to find a consensus on the impact of each physical exercise parameter in patients living with T2D.

In this review we discussed the ability of physical exercise to restore glycemic balance, compromised in patients with T2D. We can now wonder whether physical exercise could participate in T2D remission. Thus, as a remission of the disease is feasible through calorie restriction and bariatric surgery, the long-term remission of T2D has also been suggested with an intensive modification lifestyle including structured exercise [160]. T2D remission depends on various predictors such as diagnosis duration, HbA1c baseline and amount of weight loss. Thus, because different factors are involved in the heterogeneity of effects, some report the need to develop a personalized approach for managing T2D [62].

## Figures and Tables

**Figure 1 ijms-26-08182-f001:**
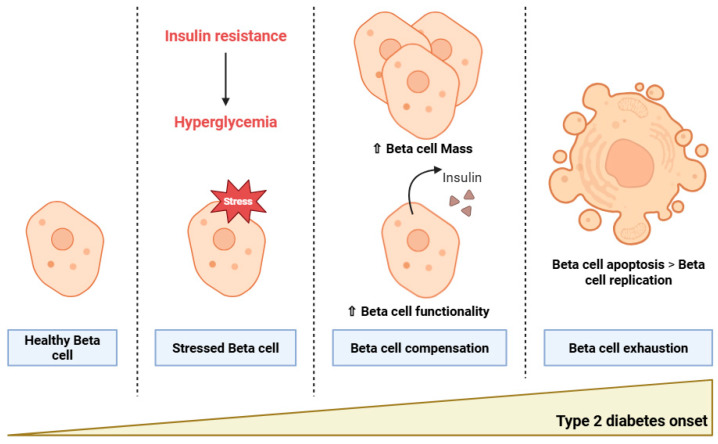
Type 2 diabetes and Beta cell dysfunction. In the context of insulin resistance, Beta cells first compensate for hyperglycemia by increasing Beta cell mass and functionality. However, when Beta cell apoptosis overtakes replication, this leads to Beta cell exhaustion. Type 2 diabetes has now emerged. Created in BioRender. Bernard, D. (2025). https://BioRender.com/ao5tr15 (accessed on 2 July 2025). White up arrows indicate an increase.

**Figure 2 ijms-26-08182-f002:**
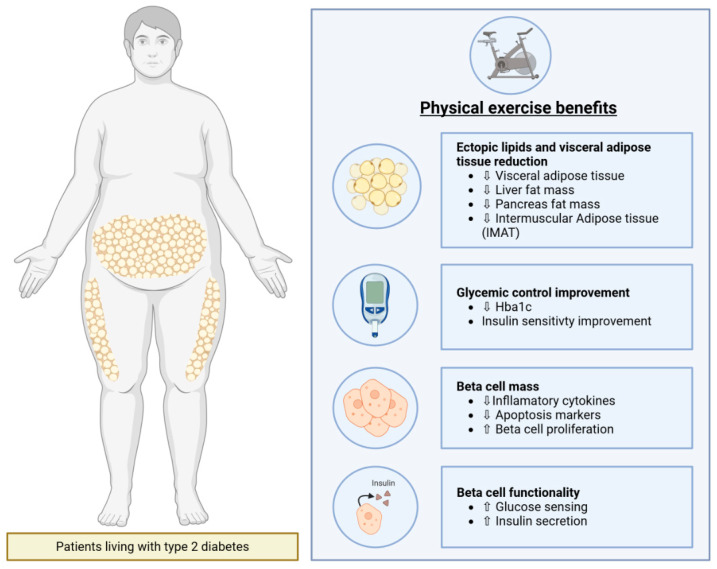
Physical exercise benefits for patients living with Type 2 diabetes. Physical exercise has many beneficial effects in patients living with Type 2 diabetes. Ectopic lipids and visceral adipose tissue reduction can contribute to overall glycemic control improvement. A decrease in hyperglycemia promotes Beta cell survival and functionality. Created in BioRender. Bernard, D. (2025) https://BioRender.com/qhke76j (accessed on 2 July 2025). White up arrows indicate an increase. White down arrows indicate a decrease.

**Figure 3 ijms-26-08182-f003:**
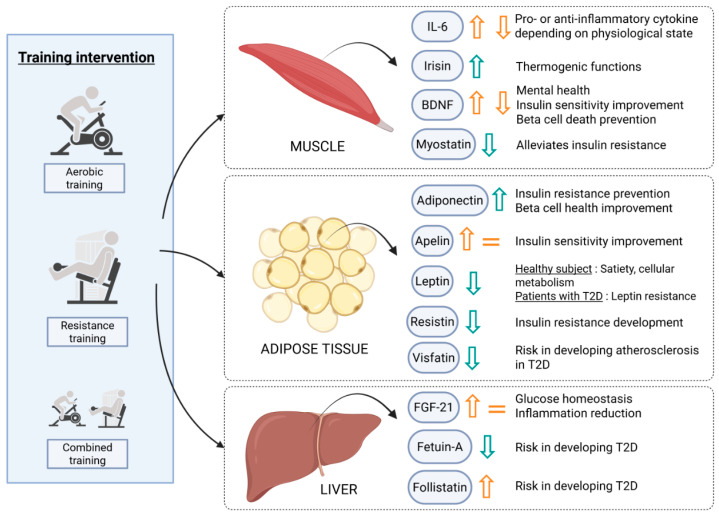
Physical exercise impacts on exerkine release and their potential effects for the management of T2D. Physical exercise is involved in exerkine secretions mainly secreted from muscles, adipose tissues, or the liver. Physical exercise enhances or decreases exerkine levels depending on the secreted molecule. (Green arrow = Accordance in scientific studies; orange arrow = more data are needed). IL-6: Interleukin-6; BDNF = Brain derived neurotrophic factor; FGF-21: Fibroblast growth factor-21. Created in BioRender. Bernard, D. (2025). https://BioRender.com/sgj2ebn (accessed on 18 august 2025).

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
