# Peer review of "Physical Exercise as a Therapeutic Approach for Patients Living with Type 2 Diabetes: Does the Explanation Reside in Exerkines?—A Review"

_ijms, 2025, doi:10.3390/ijms26178182_

Round 1

Reviewer 1 Report

Comments and Suggestions for Authors

Summary:    This review article attempts to summarize the role of “exerkines” in the treatment of type 2 diabetes.   The article is divided into 3 major sections focusing on pathophysiology of type 2 diabetes, the effect of exercise on type 2 diabetes, and the changes in “exerkines” concentrations with exercise in type 2 diabetes.      

Comments, Concerns, and Suggestions:

Given that the title and focus of the article is on “exerkines”, the Section 2 (describing the pathophysiology) and Section 3 (effect of exercise on T2D) does not seem necessary.   The section should be shortened and the section on “exerkines” expanded.  In the abstract, the authors seem to emphasize “beta cell health”, so that should be a greater focus of these sections.

Throughout the article, a greater attention to the frequency, intensity, type (aerobic, anaerobic, resistance) and duration of the exercise interventions should be made.   In some cases, these details are provided, but not in all cases.   In many instances, “physical exercise” is the only description used.

The readability of the article needs to be improved.   Most of the paragraphs within each section are summary statements of the major findings of various articles.   There is not necessary a clear main point or concluding sentence.   A reorganization and refocusing of the manuscript could help.   For example, what is the effect of moderate aerobic exercise vs. high intensity?    

Section 4 should be expanded to include a better description of each of the “exerkines”.   Where are the “exerkines” released from?   What the major effects of the “exerkines” in the normal, healthy population?   Is the release of these “exerkines” enhanced or blunted in the T2D population?  If so, what is the potential outcome?  A greater focus on the “exerkine” release and how that modifies T2D is needed.   Most of the information provides indicates whether the concentrations are higher or lower with no indication of the whether the secretions have a positive or negative impact of diabetes.

The “irisin” paragraph in Section 4.1.1 seems to focus more on the adipose tissue rather than the skeletal muscle.  

The citation formatting of many sentences is non-traditional in that the authors list the first authors entire name rather than the traditional (last name, et al.)

Author Response

Reviewer 1 :

First, we would like to thank you for your time and for all your constructive feedback which we have tried to answer as well as we can.  

Comment 1 : “Given that the title and focus of the article is on “exerkines”, the Section 2 (describing the pathophysiology) and Section 3 (effect of exercise on T2D) does not seem necessary.   The section should be shortened and the section on “exerkines” expanded.  In the abstract, the authors seem to emphasize “beta cell health”, so that should be a greater focus of these sections.”

Response 1 : We thank you for your feedback and have carefully considered your comment. We have taken into consideration your remark and tried our best to response to it. Our review intends to first (i) summarize physical exercise benefits in patients living with T2D, then (ii) suggests an underlying mechanism explaining its positive effects through exerkine secretions to finally focus on Beta cell health. We therefore feel necessary to summarize the current data available on human studies regarding the beneficial effects of physical exercise in a context of T2D (section 3), as studies in animal models are still more prevalent. Because we decided to include only protocols studying patients (human) with T2D for scientific relevance (when possible), some data are still lacking especially for studies analyzing human beta cell death.

Comment 2 : “Throughout the article, a greater attention to the frequency, intensity, type (aerobic, anaerobic, resistance) and duration of the exercise interventions should be made.   In some cases, these details are provided, but not in all cases. In many instances, “physical exercise” is the only description used.”

Response 2 : Thanks to your comment, we have been able to further clarify training intervention modalities (frequency, intensity, type, and duration) when detailed in studies.

Comment 3 : “The readability of the article needs to be improved.   Most of the paragraphs within each section are summary statements of the major findings of various articles.   There is not necessary a clear main point or concluding sentence.   A reorganization and refocusing of the manuscript could help.   For example, what is the effect of moderate aerobic exercise vs. high intensity?”

Response 3 : Thank you for your feedback. We understand that readability may be compromised because of a lack of concluding sentences. We thus have added some closing statements in particular for BDNF and Myostatin in 4.1.1 sections. In addition, we agree that studying the effect of moderate versus high intensity could be relevant. However, although the terms of the effort were specified in the review, we aimed to provide an overview of the general impact of physical exercise for the management of T2D, which can be explained by exerkines secretion. We felt necessary to include studies comparing aerobic, resistance and combined trainings. Research on exerkines is still in its early stages which may explain why conclusions cannot always be drawn.

Comment 4: “Section 4 should be expanded to include a better description of each of the “exerkines”.   Where are the “exerkines” released from?   What the major effects of the “exerkines” in the normal, healthy population?   Is the release of these “exerkines” enhanced or blunted in the T2D population?  If so, what is the potential outcome?  A greater focus on the “exerkine” release and how that modifies T2D is needed.   Most of the information provides indicates whether the concentrations are higher or lower with no indication of the whether the secretions have a positive or negative impact of diabetes.”

Response 4 : Thanks to your review, we expanded section 4 by restructured paragraphs. We have divided 4.2 section in two parts in order to further develop the impact of exerkines on beta cells. Thus, the biological functions, context of secretion/expression, the impact of T2D on exerkine levels and potential outcomes have been detailed. More information on T2D influence on irisin concentration in section 4.1.1 have also been added.

Comment 5 : “The “irisin” paragraph in Section 4.1.1 seems to focus more on the adipose tissue rather than the skeletal muscle.”

Response 5 : We first made the choice to describe the major impact of irisin which mainly takes place in adipose tissue (thermogenic functions), but we agree that secondary outcomes are also known for irisin. We thus, have added some in section 4.4.1.    

Comment 6 : “The citation formatting of many sentences is non-traditional in that the authors list the first authors entire name rather than the traditional (last name, et al.)”

Response 6 :  Citation formatting has been modified.

Reviewer 2 Report

Comments and Suggestions for Authors

This review intends to summarize the current knowledge of physical exercise benefits in a context of T2D from “unwanted” adipose tissue reduction to Beta cell health improvement. The reviewer believes that this review covers a wide range of topics, including not only glycemic control and improvement of insulin resistancec but also exerkines, and provides useful information about the potential for exercise therapy for patients with type 2 diabetes. Furthermore, the reviewer considers that this review thoroughly examines previous studies and summarizes it appropriately. However, the reviewer would like the authors to consider the following points.

Table 1 summarizes the impact of physical exercise on exerkine levels and their potential effects. However, this table does not provide basic information about the characteristics of patients and intervention programs of each study. The reviewer considers that readers would be able to understand the study better if basic information about the characteristics of patients and intervention programs of each study were added to Table 1. In particular, the reviewer thinks it important to provide whether the intervention in each study was a randomized controlled trial.

The reviewer believes that this review is significant in that it focuses not only on traditional glycemic control and improvement of insulin resistance but also on exerkin as a mechanism for the effects of physical exercise in patients with type 2 diabetes. The reviewer considers that readers’ understanding would be further enhanced by summarizing the effects of physical exercise on exerkines using illustrations as Figures 1 and 2 because this review is an appropriate summary based on the results of previous studies.

As a minor comment, most of the studies referred in this article focus on aerobic exercise. Recently, the combination of resistance exercise in addition to aerobic exercise has been recommended as exercise therapy for patients with type 2 diabetes. The reviewer thinks it would be better to mention that the physical exercise focused on in this review was primarily aerobic exercise.

Author Response

First, we would like to thank you for your positive feedback and your constructive remarks. Here are our corrections regarding your comments.

Comment 1 : “Table 1 summarizes the impact of physical exercise on exerkine levels and their potential effects. However, this table does not provide basic information about the characteristics of patients and intervention programs of each study. The reviewer considers that readers would be able to understand the study better if basic information about the characteristics of patients and intervention programs of each study were added to Table 1. In particular, the reviewer thinks it important to provide whether the intervention in each study was a randomized controlled trial.”

Response 1 : Thank you for your comment. As suggested, more information have been added to table 1 (characteristics of patients, intervention programs). Randomized controlled trials were specified when information was provided by authors.  

Comment 2 : “The reviewer believes that this review is significant in that it focuses not only on traditional glycemic control and improvement of insulin resistance but also on exerkine as a mechanism for the effects of physical exercise in patients with type 2 diabetes. The reviewer considers that readers’ understanding would be further enhanced by summarizing the effects of physical exercise on exerkines using illustrations as Figures 1 and 2 because this review is an appropriate summary based on the results of previous studies.”

Response 2 : We agree that a figure can be more appropriate to summarize the main conclusions. Thus, a figure illustrating the impact of physical exercise on exerkines and their potential effects for managing T2D has been added in page 13.

Comment 3 : “As a minor comment, most of the studies referred in this article focus on aerobic exercise. Recently, the combination of resistance exercise in addition to aerobic exercise has been recommended as exercise therapy for patients with type 2 diabetes. The reviewer thinks it would be better to mention that the physical exercise focused on in this review was primarily aerobic exercise.”

Response 3 : We agree with referee 2 and understand how you perceive the frequency with which studies involving aerobic effort appear. In our revised version we have included more references on both type of exercise. In our review, we have included all type of intervention currently presented in literrature (aerobic, resistance and combined training), and not only aerobic training, as now presented in table 1.

Round 2

Reviewer 1 Report

Comments and Suggestions for Authors

No further comments.

Reviewer 2 Report

Comments and Suggestions for Authors

I think all responses to reviewers' comments have been addressed satisfactorily.

I have no comments on the revised manuscript.